# Cell-free extracellular enzymatic activity is linked to seasonal temperature changes: a case study in the Baltic Sea

**F. Baltar[1,2,3]\*, C. Legrand[1], J. Pinhassi[1]**

[1]{Centre for Ecology and Evolution in Microbial Model Systems, Linnaeus University, Kalmar, Sweden}

[2]{Department of Marine Science, University of Otago, New Zealand}

[3]{NIWA/University of Otago Research Centre for Oceanography, Dunedin, New Zealand}

Correspondence to: F. Baltar (federico.baltar@otago.ac.nz)

## Abstract

Extracellular enzymatic activities (EEA) are a crucial step on the degradation of organic matter. Dissolved (cell-free) extracellular enzymes in seawater can make up a significant contribution of the bulk EEA. However, the factors controlling the proportion of dissolved EEA in the marine environment remain unknown. Here we studied the seasonal changes in the proportion of dissolved relative to total EEA (of alkaline phosphatase [APase], β-glucosidase, [BGase], and leucine aminopeptidase, [LAPase]), in the Baltic Sea for 18 months. The proportion of dissolved EEA ranged between 37-100%, 0-100%, 34-100% for APase, BGase and LAPase, respectively. A consistent seasonal pattern in the proportion of dissolved EEA was found among all the studied enzymes, with values up to 100% during winter and <40% during summer. A significant negative relation was found between the proportion of dissolved EEA and temperature, indicating that temperature might be a critical factor controlling the proportion of dissolved relative to total EEA in marine environments. Our results suggest a strong decoupling of hydrolysis rates from microbial dynamics in cold waters. This implies that under cold conditions, cell-free enzymes can contribute to substrate availability at large distances from the producing cell, increasing the dissociation between the hydrolysis of organic compounds and the actual microbes producing the enzymes. This might also suggest a potential effect of global warming on the hydrolysis of organic matter via a

reduction of the contribution of of cell-free enzymes to the bulk hydrolytic activity, and call for the need of further research to confirm it.

## 1    Introduction

Prokaryotes play a central role in the marine biogeochemical cycles by transforming dissolved organic matter (DOM) into living particulate organic matter (Azam and Cho, 1987). These organisms preferentially consume high molecular weight DOM, as explained by the DOM size-reactivity model (Amon and Benner, 1996; Benner and Amon, 2015). However, since only molecules <600 Da can be directly transported across the prokaryotic cell membrane (Weiss et al., 1991), heterotrophic prokaryotes need to use extracellular enzymes (EE) for hydrolyzing high molecular weight DOM into low molecular weight compounds suitable for uptake. This is why the activity of extracellular enzymatic activity (EEA) has been recognized as the initial step in organic matter degradation (Arnosti, 2011).

 EEA in aquatic environments can be cell-associated (i.e., EEs attached to the cell wall or in the periplasmatic space), or dissolved (i.e., cell-free) in the surrounding waters (Hoppe et al., 2002). Until recently, most EEA in the marine environment was believed to be associated to cells (Hoppe, 1983; Hoppe et al., 2002) leading to the perception that only cell-associated EEs were of ecological significance (Chrost and Rai, 1993; Rego et al., 1985; Someville and Billen, 1983). However, other reports suggested a major contribution of dissolved EEA to the total oceanic EEA pool (Baltar et al., 2010; Baltar et al., 2013; Duhamel et al., 2010; Karner and Rassoulzadegan, 1995; Keith and Arnosti, 2001; Obayashi and Suzuki, 2008a). This high proportion of cell-free EEA is important because it can decouple hydrolysis rates of organic material from microbial dynamics; this is, a high proportion of dissolved EEA could indicate a greater importance of the history of the water mass than of the actual processes occurring at the time of sampling (Arnosti, 2011; Baltar et al., 2010; Baltar et al., 2013; Karner and Rassoulzadegan, 1995).

Different potential sources of cell-free EEA include direct EE release from cells in response to appropriate substrate (Alderkamp et al., 2007), to bacterial starvation (Albertson et al., 1990) to changes in cell permeability (Chrost, 1991), to viral lysis (Karner and Rassoulzadegan, 1995) and to protist grazing (Bochdansky et al., 1995). However, little is known about what happens once the enzymes are free in the marine environment, including information about their lifetimes. The few available studies on the EE lifetime, indicate a

lifetime range between tens to hundreds of hours. Surface water EE lifetimes, when incubated at in situ temperature, ranged between >1 to 9 days (Bochdansky et al., 1995; Steen and Arnosti, 2011; Ziervogel and Arnosti, 2008; Ziervogel et al., 2010). However, EE lifetimes of surface waters were longer (up to 40 d) when incubated in the dark at 4°C than at the in situ conditions of light and temperature (Li et al., 1998). This is consistent with the only available study comparing cold deep versus warm surface waters EE lifetimes, where EE lifetimes were about one order of magnitude longer in the deep waters (Baltar et al., 2013). These results suggest that temperature could be a critical factor preserving the activity of cell-free EEA and thereby controlling the proportion of dissolved EEA in the marine environment.

Despite the importance and implications of cell-free EEA in marine environments, little is known about the factors that control changes in the proportion of total EEA that is dissolved. To resolve this question, a long temporal sampling strategy that accounts for the long lifetime of EEs would be desirable, because this will include the strong seasonal changes, and because a field study will be more representative of what occurs in nature than experimental manipulations, particularly when looking at seasonal temporal scales. Here we studied the temporal changes in the proportion of dissolved relative to total EEA, in a continuous biweekly sampling, with water from the Baltic Sea, for 18 months. We aimed to reveal the seasonal variability of dissolved EEA and to decipher the factors that control the proportion of dissolved relative to total EEA (of glycolytic enzymes [β-glucosidase, BGase], a proteolytic enzyme [leucine aminopeptidase, LAPase]) and alkaline phosphatase [APase]). Based on previous research suggesting longer EE lifetimes in cold environments, we hypothesized that there would be a strong link between temperature and the proportion of dissolved relative to total EEA, with lower proportions of dissolved EEA during warm periods (e.g. summer) than during cold periods (e.g. winter). This hypothesis is based on the previous evidences of higher lifetimes of EEA in cold compared to warm waters (Baltar et al., 2010), suggesting that an overall low metabolic rates of microbes would favor higher percentages of dissolved EEA because the degradation of the enzymes (i.e., microbial heterotrophic activity) is reduced under lower temperatures.

# 2 Materials and methods

## 2.1 Study site and sampling

Seawater from the Baltic Sea proper was collected twice weekly for almost 18 months, from March 22, 2012 to the August 15, 2013. Samples were taken at 2 m depth at the Linnaeus Microbial Observatory (LMO) (N 56°55.851, E 17°03.640), 10 km off the east coast of Öland, Sweden, using a Ruttner sampler. Temperature was measured on site through thermometer placed in the Ruttner sampler, and the water was transported to the laboratory in acid-washed Milli-Q-rinsed polycarbonate bottles within 1 h.

## 2.2 Biotic and abiotic environmental parameters

Chlorophyll a (Chl *a*) concentration was analyzed following extraction using ethanol (Jespersen and Christoffersen, 1987). Chlorophyll a (Chl a) concentration, dissolved inorganic nutrients ($NH_4^+$, $NO_3^-$, $PO_4^{3-}$ and $SiO_2$) and were analysed following previously described protocols (Jespersen and Christoffersen, 1987; Valderrama, 1981). Dissolved organic carbon (DOC) was measured via high-temperature catalytic oxidation using a TOC-V detector coupled to a TNM-1 unit (Shimadzu Corporation) (Pages and Gadel, 1990). Bacterial abundance was determined using flow cytometry according to the protocol described in Del Giorgio et al. (1996). Bacterial heterotrophic production was derived from $^3$H-leucine incorporation rates measured on quadruplicates and two controls according to Smith and Azam (1992).

## 2.3 Measurement of total and dissolved extracellular enzymatic activity (EEA)

The hydrolysis of the fluorogenic substrate analogues 4-methylcoumarinyl-7-amide (MCA)-L-leucine-7-amido-4-methylcoumarin, 4-methylumbelliferyl (MUF)-β-D-glucoside and MUF-phosphate was analyzed to estimate potential activity rates of leucine aminopeptidase (LAPase), β-glucosidase (BGase), and alkaline phosphatase (APase), respectively (Hoppe, 1983). The procedure was followed as previously described (Baltar et al., 2010; Baltar et al., 2013; Baltar et al., 2009). Briefly, EEA was determined after substrate addition and incubation using a spectrofluorometer with a microwell plate reader (FLUOstar – BMG Labtech) at excitation and emission wavelengths of 365 and 445 nm, respectively. Samples

(300 µl) were incubated in the dark at *in situ* temperature for 1.5-3 h. Subsamples without
substrate additions served as blanks to determine the background fluorescence of the samples.
Previous experiments showed insignificant abiotic hydrolysis of the substrates (Azúa et al.,
2003; Hoppe, 1993; Unanue et al., 1999). The fluorescence obtained at the beginning and the
end of the incubation was corrected for the corresponding blank. The increase in fluorescence
over time was transformed into hydrolysis activity using a standard curve established with
different concentrations of the fluorochromes MUF and MCA added to 0.2 µm filtered sample
water. A final substrate concentration of 31.2 µmol $l^{-1}$ was used to measure BGase activities,
100 µmol $l^{-1}$ for APase and 500 µmol $l^{-1}$ for LAPase. These concentrations were previously
determined as saturating substrate concentrations.
The total and the dissolved fraction of the EEA was distinguished as previously described
(Baltar et al., 2010; Baltar et al., 2013). Briefly, raw seawater was used for total EEA;
whereas for dissolved EEA, samples were gently filtered through a low protein-binding 0.2
µm Acrodisc Syringe filter (Pall) for dissolved EEA following the protocol of (Kim et al.,
2007). The use of low protein-binding filters is important in this context since the adsorption
of extracellular enzymes depends on the type of the filter material used for size fractionation
(Obayashi and Suzuki, 2008a). In the present study, dissolved (cell-free) EEA is defined as
the EEA recovered in the filtrate. Total and dissolved EEA were determined on six replicate
samples each.
**2.4   Statistical analyses**
The relations between variables were examined by means of correlation analysis computing
Pearson pairwise statistics. Normality was checked with a Shapiro-Wilks test before Pearson
correlations were calculated.

**3   Results and discussion**
A clear seasonal pattern in temperature was observed, with lower and relatively stable
temperatures during winter (3-4°C), and strong increases during spring-summer (up to 20°C),
followed by a quick temperature drop in autumn (Fig. 1A). Chlorophyll-a concentration
varied between 0.4 to 4.8 µg $l^{-1}$, with maximum peaks during the two types of blooms that
typically occur in the Baltic (Lindh et al., 2015), the diatom and dinoflagellate spring bloom
(April-May) and the cyanobacterial summer bloom (July-September) (Legrand et al., 2015)
(Fig. 1B).
The total (bulk) EEA of APase, LAPase and BGase followed a similar temporal pattern (Fig.
2A). APase was the EEA with the highest rates (ranging from 1.5-32 nmol $l^{-1}$ $h^{-1}$), followed
by LAPase (from 0.6-9.3 nmol $l^{-1}$ $h^{-1}$) and BGase (from 0.1-21 nmol $l^{-1}$ $h^{-1}$), indicating a
potential significant P limitation in the Baltic Sea (Granéli et al., 1990; Hagström et al., 2001).
The strongest peaks in BGase and LAPase co-occurred (June and August 2012, May and
August 2013), and these enzymes were significantly correlated (Pearson's r = 0.49, $p$=0.024).
These peaks in BGase and LPase coincided with some of the APase and the Chla-a peaks,
suggesting a potential link between these different enzymes and the phytoplankton dynamics.
APase was significantly correlated to BGase (Pearson's r = 0.53, $p$=0.013) but not to LAPase
(Pearson's r = 0.20, $p$=0.346). The BGase:LAPase ratio, which have been generally suggested
to be indicative of the relative degradation of polysaccharides relative to proteinaceous
material, peaked when the highest BGase and LAPase rates were observed (May-June and
August-October), just following after the diatom/dinoflagellate spring bloom (April-May) and
the cyanobacterial summer blooms (July-September). This is agreement with the results
obtained in a recent seasonal study in the Adriatic Sea, where BGase prevailed over LAPase
associated to phytoplankton blooms (Celussi and Del Negro, 2012). Nevertheless, results
from other reports question the validity of the BGase:LPase ratio as an indicator of the
relative degradation of polysaccharides relative to proteins. Previous investigations of a range
of peptidases substrates (Bong et al., 2013; Obayashi and Suzuki, 2008b, 2005) have shown
that LAPase activity levels vary in a manner not indicative of other peptidase activities, and
other recent report suggested that LAPase should not be interpreted as a quantitative proxy of
the total peptidolytic potential of the community (Steen et al., 2015). Furthermore,
measurements of BGase do not present a complete or representative picture of the broad and
variable spectrum of polysaccharide hydrolases present in the ocean (Arnosti, 2011). There
was a tendency for higher EEA rates (and BGase:LAPase ratio) in 2013 as compared to 2012,
likely explained by the reported interannual variability of phytoplankton communities linked
to environmental conditions  in the Baltic Sea (Kahru and Elmgren, 2014; Legrand et al.,
2015), since different phytoplankton groups can release diverse types of organic carbon
compounds, which would likely select for different bacterioplankton groups/enzymes
(Pinhassi et al., 2004).
The proportion of dissolved relative to total EEA ranged between 0-100%, where LAPase and
APase showed a similar range (37-100% and 37-100%, respectively) and BGase showed the
broadest range (0-100%) (Fig. 3). LAPase was the EEA with the lowest seasonal amplitude
variability in the proportion of dissolved EEA, wheras BGase showed the largest seasonal
variability. These values are within the same ranges reported in the surface coastal North Sea
waters (Someville and Billen, 1983), Tokyo Bay (Hashimoto et al., 1985), Mediterranean Sea
(Karner and Rassoulzadegan, 1995), Elbe estuary (Karrasch et al., 2003), Gulf of Mexico
(Ziervogel and Arnosti, 2008; Ziervogel et al., 2010), North Pacific Subtropical Gyre
(Duhamel et al., 2010) and the epipelagic to bathypelagic waters of the Atlantic (Baltar et al.,
2010; Baltar et al., 2013).
The proportion of dissolved EEA showed a clear seasonal pattern in our study (despite the
great variability in the bulk EEA rates), with higher values in winter and a pronounced
decrease in summer. Overimposed on this seasonal pattern, there were sometimes of the year
(March-May) when there were stronger fluctuations in the proportion of dissolved EEA (Fig.
3). This coincides with the phytoplankton bloom as indicated by the increases in Chl-a
concentration (Fig. 1B) and in APase (Fig. 2A); which is consistent with the rapid succession
observed by next generation sequencing in different phytoplankton (and bacterioplankton)
taxa during those months in this study site (Lindh et al., 2015).
This overall seasonal pattern in the % of dissolved EEA was conserved among all the
enzymes studied (APase, BGase and LAPase), suggesting that the main factors regulating the
proportion of dissolved EEA affect all enzymes equally, irrespectively of their metabolic
function. This advocates for some environmental factor rather than a biological factor
controlling the proportion of dissolved EEA. The most logical factors that could be
responsible for this seasonal pattern would be light and/or temperature. Inactivation of
extracellular enzymes by photochemical reactions have been found in biofilm microbiota
(Espeland and Wetzel, 2001). However, in laboratory experiments with Arctic surface waters,
the activity of cell-free EEs (APase and LAPase) were not affected by light (under full
spectrum natural sunlight), suggesting that photochemical reactions are not relevant pathway
for the decay of cell-free EE in seawater (Steen and Arnosti, 2011). Indeed, we found that
only temperature was significantly correlated to the proportion of dissolved EEA of the three
enzymes studied (APase (Pearson's r = -0.60, $p$=0.0035)), BGase (Pearson's r = -0.73,
$p$=0.0002) and LAPase (Pearson's r = -0.68, $p$=0.0006)) (Table 1). This strong correlation

between temperature and the proportion of dissolved EEA was always negative, suggesting that lower temperatures favour the proportion of cell-free EEA. These results are consistent with the negative effect of temperature on EE lifetimes found in an incubation experiment with Red Sea surface water (Li et al., 1998), the extended life time of cell-free EE in Arctic waters (Steen and Arnosti, 2011), and with the order of magnitude higher EE lifetimes found in the deep as compared to the surface waters of the Atlantic (Baltar et al., 2013). This suggests that low temperature preserves better (than warm temperature) the constitutive activity of the cell-free enzymes, allowing them to remain active for longer periods. This might be linked to a reduction in the metabolism of heterotrophic microbes that would reduce the consumption/degradation rates of dissolved EEs. This hypothesis is supported by the significant negative correlations found between the proportion of dissolved BGase and LAPase and the bacterial heterotrophic production and DOC concentration (Table 1), which suggest that the proportion of dissolved EEA (BGase and LAPase) is higher when the microbial heterotrophic degrading activities are lower (during winter). This decreased heterotrophic activity would facilitate the preservation or accumulation of dissolved EEA. The higher proportion of dissolved EEA during winter suggests that the decoupling of *in situ* hydrolysis rates from actual microbial dynamics is stronger during winter (or in cold waters). Thus, these results underpin the importance of considering this stronger cold-related decoupling when relating EEA to other microbial processes. These results also suggest that, under the projected global warming scenario, it could be possible that the hydrolysis of organic matter due to cell-free EE might be reduced due to a shorter lifetime of the EEs.

## 4   Conclusions

Overall, the results of this study suggest that a relevant fraction of the total EEA measured in a particular environment can be due to free EEs, which might be a consequence of the substrate history of the water masses. Thus, advection of dissolved EEA might be a critical source of EEA at any given environment (Baltar et al., 2010; Baltar et al., 2013; Steen and Arnosti, 2011). Other factors might also allow for extended EE lifetimes during advection, like association of EEs to particles (Gianfreda and Scarfi, 1991; Naidja et al., 2000; Ziervogel et al., 2007), to exopolymeric matrix (Decho, 1990), and to detrital particle complexes (Nagata and Kirchman, 1992). Low temperature seems to be a critical factor favoring longer EE lifetimes and thereby higher proportions of dissolved relative to total EEA. This implies

that under cold conditions, cell-free enzymes can contribute to substrate availability at large distances from the producing cell, potentiating the disconnection between the hydrolysis of organic compounds and the actual microbes producing the enzymes. Moreover, under warmer conditions, like those predicted to occur due to global warming, the hydrolysis of organic matter (i.e., rate limiting step in the degradation of organic matter) can be reduced due to a lower contribution of the cell-free EE hydrolysis.

## Acknowledgements

We acknowledge Anders Månsson, Emil Fridolfsson, Kristofer Bergström, Saraladevi Muthusamy, Markus Lindh, Mireia Bertos Fortis, Oscar Nordahl for their never-ending energy to carry out field sampling and Sabina Arnautovic and Emmelie Nilsson for their skillful technical assistance in the processing of samples. This research was supported by grants from the European Science Foundation EuroEEFG project MOCA and The Swedish Research Council (VR) to JP, and the Strategic Marine Environmental Research program ECOCHANGE funded by the Swedish Research Council (Formas) to JP and CL. FB was supported by a University of Otago Research Grant. We would like to acknowledge the support and insightful comments of the reviewers, which clearly helped improve the overall merit of the manuscript.

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

Table 1. Parson's correlation coefficient (r) between proportion of dissolved extracellular
enzymatic activities (alkaline phosphatase [APase], β-glucosidase, [BGase], and leucine
aminopeptidase [LAPase]) and Chlorophyll-a, salinity, inorganic nutrients, dissolved organic
carbon (DOC), bacterial abundance (BA) and bacterial heterotrophic production (BP). Values
of r are significant at $p < 0.05$ are highlighted in bold.

| | % APase | | %BGase | | %LAPase | |
|---|---|---|---|---|---|---|
| | r | p-value | r | p-value | r | p-value |
| Temp | **-0.60** | **0.004** | **-0.73** | **0.001** | **-0.68** | **0.001** |
| Chlorophyll-a | -0.16 | 0.499 | -0.43 | 0.052 | **-0.57** | **0.006** |
| Salinity | -0.22 | 0.330 | 0.18 | 0.443 | 0.42 | 0.059 |
| Nitrate | 0.21 | 0.372 | 0.34 | 0.132 | **0.46** | **0.034** |
| Phosphate | 0.38 | 0.088 | **0.47** | **0.030** | **0.51** | **0.017** |
| Silicate | 0.27 | 0.243 | 0.20 | 0.379 | 0.24 | 0.288 |
| Ammonium | 0.03 | 0.892 | -0.07 | 0.758 | -0.04 | 0.862 |
| DOC | -0.18 | 0.433 | **-0.66** | **0.001** | **-0.65** | **0.001** |
| BA | -0.32 | 0.157 | -0.39 | 0.077 | -0.16 | 0.500 |
| BP | -0.38 | 0.092 | **-0.64** | **0.002** | **-0.63** | **0.002** |

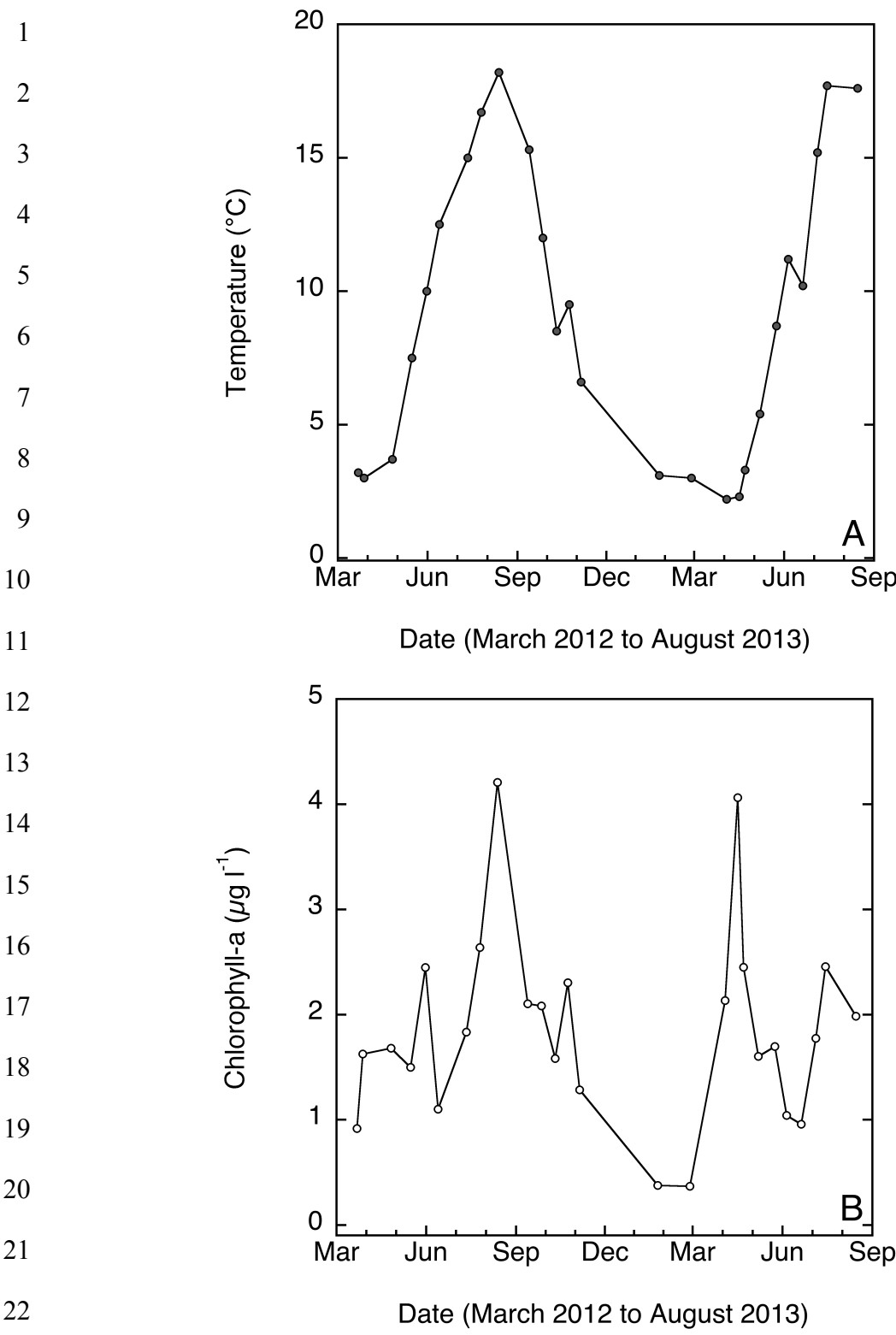

24 Figure 1. Temporal dynamics in (A) temperature and (B) chlorophyll-a concentrations, at the

25 Linnaeus Microbial Observatory (LMO, Baltic Sea) from March 2012 to August 2013

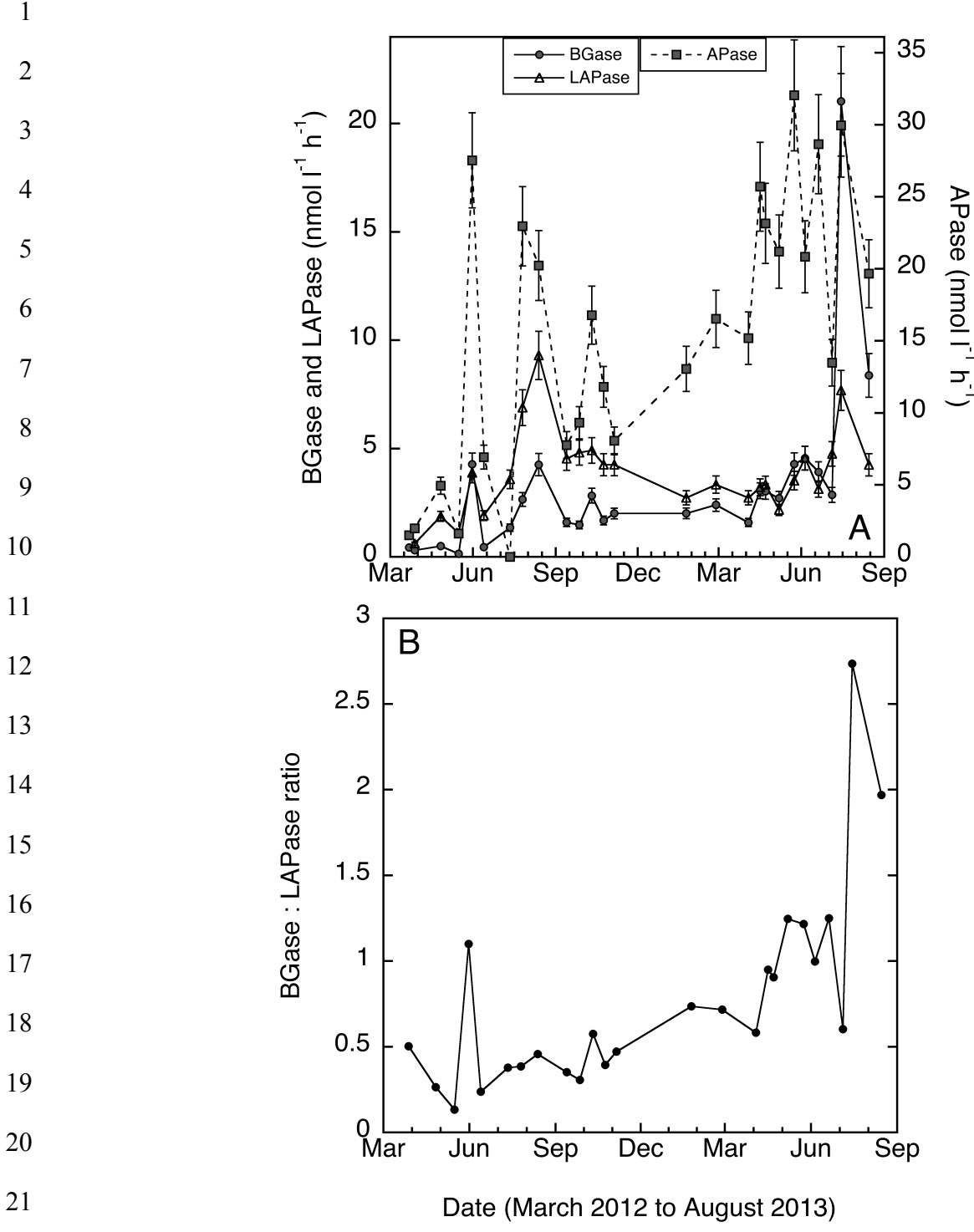

Figure 2. Temporal dynamics in (A) extracellular enzymatic activities of alkaline phosphatase (APase), β-glucosidase, (BGase), and leucine aminopeptidase, (LAPase) and), and (B) the BGase:LAPase ratio, at the Linnaeus Microbial Observatory (LMO, Baltic Sea) from March 2012 to August 2013.,

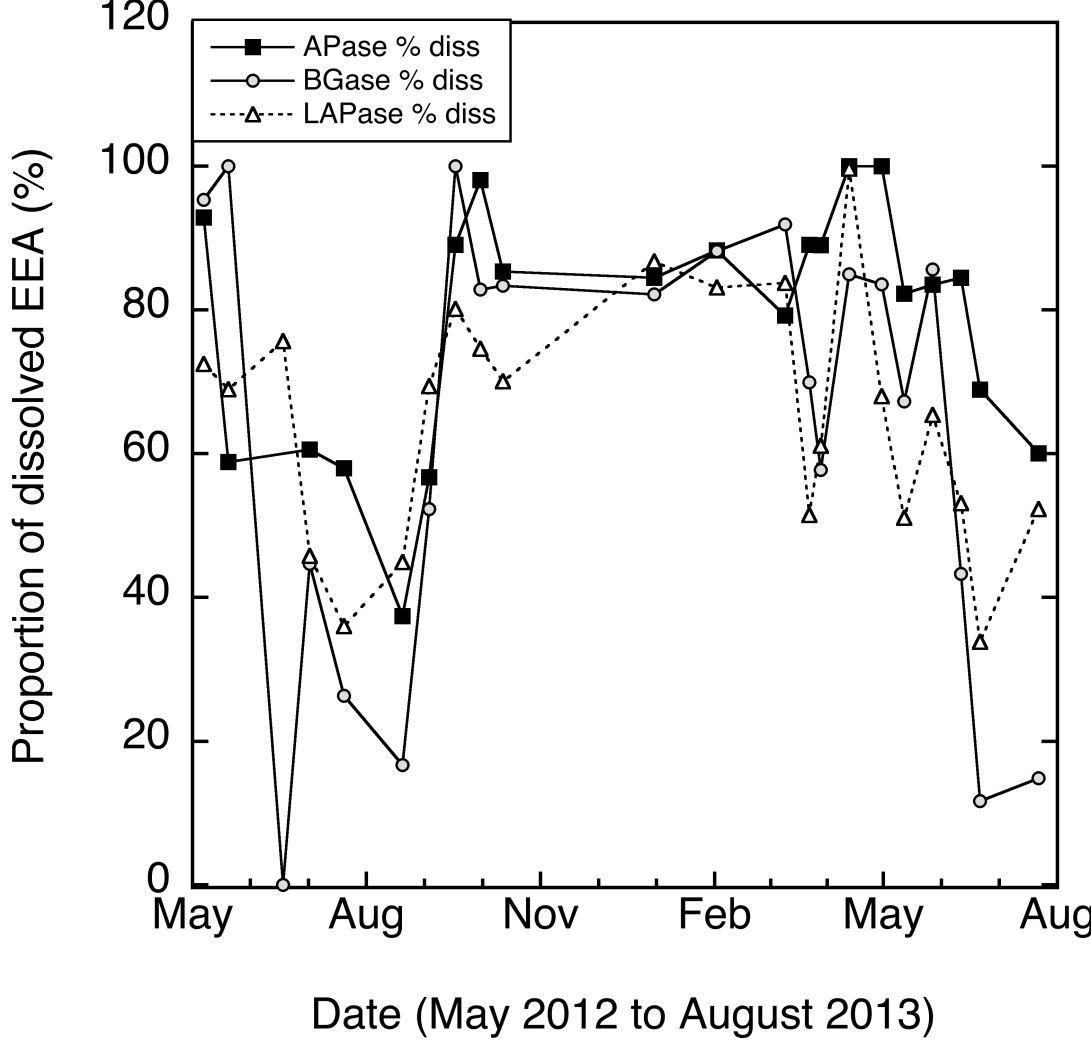

Figure 3. Temporal dynamics in the proportion of dissolved relative to total extracellular
enzymatic activities (EEA) of alkaline phosphatase (APase), β-glucosidase, (BGase), and
leucine aminopeptidase, (LAPase), at the Linnaeus Microbial Observatory (LMO, Baltic Sea)
from May 2012 to August 2013.

