# Peer review of "Cell-free extracellular enzymatic activity is linked to"

_Biogeosciences, 2015_

## Referee Comment (RC1) · Anonymous Referee #1 · 24 Jan 2016

General comment The article by Baltar et al. shows a strong relationship between the proportion of dissolved extracellular enzymatic activities and seasonal variations (temperature) occurring at a coastal site in the Baltic Sea. The topic of the manuscript is very important, and not fully understood yet. It is a clear and straightforward document that describes a valid experimental design. My only concern regards the lenght of the paper. In particular, the authors focus only on one important environmental factor: temperature. In order to make the article's conclusions sound (e.g. the potential effect of global warming on organic matter degradation performed by cell-free enzymes) some experimental manipulations could have been performed. Alternatively, the effect of factors other than temperature could have been considered (and eventually discarded if

not significant). For these reasons I suggest to improve the analysis with other environmental factors (if available). Alternatively, the manuscript should be considered for a 'short note' type of paper. Since I'm aware that BG does not include 'short notes' as potential article types, an option could be the 'Ideas and perspectives' type, within which the article could perfectly fit in its current form with a slight modification of the title (just as a suggestion, it could become something like: "temperature affects the activity of cell-free extracellular enzymes: a seasonal case study in the Baltic Sea).

Specific comments:

Page 1 line 27-29: This conclusive sentence is very strong: although potentially true it would deserve further investigations (see General comment) Page 2 line 3: although heterotrophic prokaryotes are much of the story, I would not ascribe only to them the pivotal DOM reworking role, especially when EEA are involved (exoenzymes from cyanobacteria can be even more efficient). I'd be more cautious just saying 'prokaryotes'. Page 2 line 15: extra ')' Page 3 line 25: "Temperature was measured on site" How? Was water collected and temperature measured on a aliquot by means of a thermometer? Anyhow this should be indicated. Page 6 line 8: I am afraid this (temperature and light) is a little restrictive. pH is an important factor in enzyme activity, especially in extracellular ones. What is the pH variability in the 'low-salinity' high-chlorophyll Baltic sea? I suspect that in the sampling area pH variations might be broader that those tested in many ocean acidification experiments for which many references are available. Substrate concentration and composition also affect enzyme activity with or without links to the metabolic state of the source organism (Arnosti 2011) Catalytic elements have also been shown to drive (to some extent) hydrolysis rates of at least LAPase (Fukuda et al., 2000). These aspects (especially pH and substrate) should be mentioned. Page 10 line 30: heterotrophic MARINE flagellates Figure 2, upper panel: error bars should be shown
* * *

---

## Referee Comment (RC2) · Anonymous Referee #2 · 26 Mar 2016

General comments: The authors attempt to address an interesting and important problem, with a time series of measurements of potential enzyme activities, including the fraction that is cell free. These measurements are a good start, but the data lack context, and the framework which the authors build does not adequately reflect current understanding of microbial community activities and dynamics.

Are other data in addition to T and chl a available? Salinity, DOC, circulation, other physical or chemical data for this site? A multitude of factors can affect potential enzyme activities, as well as the fraction of activity that is cell-free. Are cell counts available, or are there other studies done at the same location that provide a more in-depth background on the microbial ecology of the system? In particular, the Baltic has a

<printer-friendly version>

very strong N-S salinity gradient that is linked with both compositional and functional changes in microbial communities (e.g. Dupont et al. (2014). "Functional tradeoffs underpin salinity-driven divergence in microbial community composition." PLOS one 9: 89549). It would be helpful to have some sense of microbial dynamics at this site; the T and chl a data are not really a sufficient context.

In its current form, this manuscript lacks sufficient significance and quality and presentation; it needs a lot of work.

Specific comments: Pg 2 lines 11, 12 Note that Arnosti (2011) is cited improperly ("This is why the activity of extracellular enzymatic activity (EEA) has been recognized as the rate-limiting step in organic matter degradation (Arnosti, 2011)." Arnosti (2011) does not state that EEA is the rate limiting step: that reference points out that EEA is the initial step in organic matter degradation, and discusses this idea at great length. These statements are not the same!

Lines 18, 19: earlier reports of cell-free activity, including Karner & Rassoulzadegan (1995), Obayashi & Suzuki (2008) and Keith & Arnosti (2001).

Note that one of the key points of the current ms - that enzyme activities may be decoupled from the producing organisms, and may contribute greatly to carbon cycling - is explicitly also discussed in Arnosti (2011), and summarized as one of the key points in that manuscript.

Lines 31 forward: Steen & Arnosti (2011) investigated cell-free enzyme lifetime in Arctic surface waters

Pg. 3: Line 7, delete 'the' before 'cell-free' Line 8, delete 'what are' before 'the factors' Line 8, this thought is incomplete. Do you mean 'little is known about the factors that control changes in the proportion of total EEA that is dissolved' ? Line 9, move 'is needed' to the end of the sentence Line 11: move the information about the specific enzyme activities assayed into another sentence following this one. Line 14: change

'decipher what factors' to 'decipher the factors that control' Line 16, change to 'we hypothesized that there would be a strong link', or something of this sort- need to improve the phrasing.

Line 7 forward. This paragraph is not very clear – why will a long-term sampling strategy in particular help resolve the question of when and under what conditions cell-free EE is important? One could easily imagine that a detailed biochemical investigation of the nature and structure and dynamics of cell-free enzymes is what is needed (tho this of course is technically very challenging, and is a different question than the one examined here.) The whole paragraph needs more focus.

As it stands currently, this hypothesis is not very well phrased. At minimum, add another sentence or two: is this hypothesis based on rates of molecular motion, on rates of diffusion of enzymes and proteases, on an idea of the structural basis of enzymes that are produced under cold vs warm conditions, on kinetics of degradation/inactivation reactions?

Pg. 4 Were killed controls (autoclaved water) subtracted from the measured values?

Are other data (salinity, for example) available for this site? How variable are cell counts, or bacterial productivity, or DOM or POM or other biogeochemical parameters for this site?

Pg 5: The data presentation and discussion on pg 5 is not very satisfactory. The BGase and LAPase plots do appear to track one another to some extent, but having the APase activity on the same plot and scale makes something of a mess. Try plotting with a secondary Y axis for APase. More critically, it is difficult to figure out what connections there might be: as the text says, there are sometimes –but not always- coincidental peaks in these different activities, but this pattern is not very strong.

Moreover, the authors assert that BGase and LAPase show relative protein/polysaccharide degradation ratios. Note that more recent work has demonstrated

that this perspective is likely limited. Multiple investigations of a range of peptidases substrates (see in particular the work of Obayashi & Suzuki Obayashi, Y. and S. Suzuki (2005). "Proteolytic enzymes in coastal surface seawater: Significant activity of endopeptidases and exopeptidases." Limnol. Oceanogr. 50: 722-726. Obayashi, Y. and S. Suzuki (2008). "Occurrence of exo- and endopeptidases in dissolved and particulate fractions of coastal seawater." Aq. Microb. Ecol. 50: 231-237. Bong, C. W., Y. Obayashi and S. Suzuki (2013). "Succession of protease activity in seawater and bacterial isolates during starvation in a mesocosm experiment." Aq. Microb. Ecol. 69: 33-46) which demonstrates that LAPase activity levels vary in a manner not indicative of other peptidase activities. See also the recent work of Steen et al. (2015; "Substrate specificity of aquatic extracellular peptidases assessed by competitive inhibition assays using synthetic substrates" AME 75: 271-281), who investigate the specificity of LAPase-hydrolyzing enzymes, and comment that "In some studies, Vmax for Leu-AMC hydrolysis is interpreted as a quantitative proxy of the total peptidolytic potential of a community (Kellogg et al. 2011). The results presented here show that approach to be invalid." Furthermore, the work of Arnosti (2011), which is cited in the ms, discusses the issue of the broad and variable spectrum of polysaccharide hydrolases present in the ocean; measurements of BGase do not present a complete or representative picture.

Note in addition that the peaks in BGase/LAPase ratios (which are not all that convincing..this plot shows a general increasing trend over the timecourse of the study, and the single large peak is driven by 2 measurements of BGase activity) do not in fact coincide with the chl a max, since the peak chl a in Fig 1B occurs in ca August and then in May, whereas Fig 2b shows peaks in June and August.

The comments on lines 19-22 on different phytoplankton blooms and enzyme activities is simply not very convincing: APase activity might be directly connected to phytoplankton, given the presence of APase in some phytoplankton, but the BGase and LAPase connection would be more distant, given that heterotrophic prokaryotes are the sources

of these enzymes.

The discussion of all of these data in fact leave the reader with the impression that the authors don't quite know what to do with this long time series. More digging in the literature, discussion with other colleagues who have data from this site might help, but this is currently a very weak part of the manuscript.

Pg 6 The high fraction of dissolved activity is indeed an interesting observation. The authors need to discuss here some ideas about why this might be - certainly others have also at times high cell-free activities, but these activities are high for most of the annual cycles. Moreover, the changes in the cell-free activities for example between March and May, when the data spikes up and down with considerable frequency, would allow for some discussion of lifetimes (for activity to drop this much, some of the enzymes must either be removed, or they are advected away, and one is sampling a different patch of water; to evaluate this, more data on the study site and the dynamics of the water masses is needed.)

As others have doubtless pointed out, correlation is not causation: there is a statistical correlation between T and cell-free activity, but this is a long way from a coherent or plausible explanation. To what extent do cell counts, bacterial productivity, or microbial community composition change at this site on seasonal scales? What is the physical oceanography of this setting –what water masses are sampled? It is extremely likely that the nature and types of enzymes that hydrolyze leu-MCA, B-MUF, and alkaline phosphatase are likely quite different under different conditions of productivity, and with different seasons, all factors that are correlated with differences in microbial community composition and activity. See for example Arrieta & Herndl (2002; "Changes in $\beta$-glucosidase diversity during a coastal phytoplankton bloom." Limnol Oceanogr 47: $594-599$). In any case, drawing parallels between community activities in the Arctic and the Baltic needs also to consider the differences between permanently cold and temperate environments, in terms of community composition and activities.

With all of these issues that should be addressed, extending the discussion here to questions of global warming is much too far a reach.

---

## Author Comment (AC1) · 30 Mar 2016

We thank the reviewer for the constructive comments on this manuscript. We have taken them on board and our responses to reviewer comments, including potential modifications to the manuscript, are detailed in the following:

REVIEWER COMMENT 1 by Referee #1: General comment The article by Baltar et al. shows a strong relationship between the proportion of dissolved extracellular enzymatic activities and seasonal variations (temperature) occurring at a coastal site in the Baltic Sea. The topic of the manuscript is very important, and not fully understood yet. It is a clear and straightforward document that describes a valid experimental design.

Author response: We appreciate the positive comments of the reviewer

REVIEWER COMMENT 2 by Referee #1: My only concern regards the lenght of the paper. In particular, the authors focus only on one important environmental factor: temperature. In order to make the article's conclusions sound (e.g. the potential effect of global warming on organic matter degradation performed by cell-free enzymes) some experimental manipulations could have been performed. Alternatively, the effect of factors other than temperature could have been considered (and eventually discarded if not significant). For these reasons I suggest to improve the analysis with other environmental factors (if available). Alternatively, the manuscript should be considered for a 'short note' type of paper. Since I'm aware that BG does not include 'short notes' as potential article types, an option could be the 'Ideas and perspectives' type, within which the article could perfectly fit in its current form with a slight modification of the title (just as a suggestion, it could become something like: "temperature affects the activity of cell-free extracellular enzymes: a seasonal case study in the Baltic Sea).

Author response: We will include other factors than temperature and light (e.g. nutrients, salinity, bacterial abundance, bacterial production) to the analysis and text in the revised manuscript. We appreciate the title suggested by the reviewer, and will use it in the revised version.

REVIEWER COMMENT 3 by Referee #1: Specific comments: Page 1 line 27-29: This conclusive sentence is very strong: although potentially true it would deserve further investigations (see General comment)

Author response: We will rewrite this sentence to make it less strong and to include a suggestion for the need for further investigations in the revised version.

REVIEWER COMMENT 4 by Referee #1: Page 2 line 3: although heterotrophic prokaryotes are much of the story, I would not ascribe only to them the pivotal DOM reworking role, especially when EEA are involved (exoenzymes from cyanobacteria can be even more efficient). I'd be more cautious just saying 'prokary- otes'. Page 2 line

15: extra ')'

Author response: We will modify this in the revised version.

REVIEWER COMMENT 5 by Referee #1: Page 3 line 25: "Temperature was measured on site" How? Was water collected and temperature measured on a aliquot by means of a thermometer? Anyhow this should be indicated.

Author response: Water was collected with Ruttner sampler equipped with a termometer inside. Temperature was measured on site, through termometer placed in the sampler. We will include this information in the revised version.

REVIEWER COMMENT 6 by Referee #1: Page 6 line 8: I am afraid this (temperature and light) is a little restrictive. pH is an important factor in enzyme activity, especially in extracellular ones. What is the pH variability in the 'low-salinity' high-chlorophyll Baltic sea? I suspect that in the sampling area pH variations might be broader that those tested in many ocean acidification experiments for which many references are available. Substrate concentration and composition also affect enzyme activity with or without links to the metabolic state of the source organism (Arnosti 2011) Catalytic elements have also been shown to drive (to some extent) hydrolysis rates of at least LAPase (Fukuda et al., 2000). These aspects (especially pH and substrate) should be mentioned.

Author response: As mentioned in the response to the reviewer comment 2 we will include other parameters as well, however we did not measure pH. Nevertheless, the strongest variations in pH would be caused by autotrophic and heterotrophic activity, and we are including phytoplankton biomass (chlorophyll-a) and will include bacterial heterotrophic activity, which will be indicative of fluctuations in pH.

REVIEWER COMMENT 7 by Referee #1: Page 10 line 30: heterotrophic MARINE flagellates

Author response: We will correct this in the revised version of the ms.

REVIEWER COMMENT 8 by Referee #1: Figure 2, upper panel: error bars should be shown

Author response: We will add the error bars to Figure 2 in the revised version.

---

## Author Comment (AC2) · 30 Mar 2016

We thank the reviewer for the constructive comments on this manuscript. We have taken them on board and our responses to reviewer comments, including potential modifications to the manuscript, are detailed in the following:

REVIEWER COMMENT 1 by Referee #2: General comments: The authors attempt to address an interesting and important prob- lem, with a time series of measurements of potential enzyme activities, including the fraction that is cell free. These measurements are a good start, but the data lack con- text, and the framework which the authors build does not adequately reflect current understanding of microbial community activities and dynamics.

Author response: We will increase the data context by including additional variables, and adjust the framework according to the reviewer's comments, in the revised version of the ms.

REVIEWER COMMENT 2 by Referee #2: Are other data in addition to T and chl a available? Salinity, DOC, circulation, other physical or chemical data for this site? A multitude of factors can affect potential en- zyme activities, as well as the fraction of activity that is cell-free. Are cell counts avail- able, or are there other studies done at the same location that provide a more in-depth background on the microbial ecology of the system? In particular, the Baltic has a very strong N-S salinity gradient that is linked with both compositional and functional changes in microbial communities (e.g. Dupont et al. (2014). "Functional tradeoffs underpin salinity-driven divergence in microbial community composition." PLOS one 9: 89549). It would be helpful to have some sense of microbial dynamics at this site; the T and chl a data are not really a sufficient context. In its current form, this manuscript lacks sufficient significance and quality and presentation; it needs a lot of work.

Author response: Yes, we have other data available, which we will include in the revised version (Salinity, DOC, etc.).

REVIEWER COMMENT 3 by Referee #2: Specific comments: Pg 2 lines 11, 12 Note that Arnosti (2011) is cited improperly ("This is why the activity of extracellular enzy- matic activity (EEA) has been recognized as the rate-limiting step in organic matter degradation (Arnosti, 2011)." Arnosti (2011) does not state that EEA is the rate limiting step: that reference points out that EEA is the initial step in organic matter degradation, and discusses this idea at great length. These statements are not the same!

Author response: We are sorry for this misunderstanding, this will be corrected accordingly in the revised version.

REVIEWER COMMENT 4 by Referee #2: Lines 18, 19: earlier reports of cell-free activity, including Karner & Rassoulzadegan (1995), Obayashi & Suzuki (2008) and

Keith & Arnosti (2001). Note that one of the key points of the current ms - that enzyme activities may be decoupled from the producing organisms, and may contribute greatly to carbon cycling - is explicitly also discussed in Arnosti (2011), and summarized as one of the key points in that manuscript.

Author response: Karner & Rassoulzadegan (1995) and Obayashi & Suzuki (2008) are cited in the manuscript, and we will also include Keith & Arnosti (2001) in the revised version. We will include a specific mention of the previous papers where the decoupling of EEA from the producing organisms has been suggested before, including Arnosti (2011) and Baltar et al. (2010).

REVIEWER COMMENT 5 by Referee #2: Lines 31 forward: Steen & Arnosti (2011) investigated cell-free enzyme lifetime in Arctic surface waters

Author response: we will include that reference in that specific statement as well in the revised version of the ms.

REVIEWER COMMENT 6 by Referee #2: Pg. 3: Line 7, delete 'the' before 'cell-free' Line 8, delete 'what are' before 'the factors' Line 8, this thought is incomplete. Do you mean 'little is known about the factors that control changes in the proportion of total EEA that is dissolved' ? Line 9, move 'is needed' to the end of the sentence Line 11: move the information about the specific enzyme activities assayed into another sentence following this one. Line 14: change 'decipher what factors' to 'decipher the factors that control' Line 16, change to 'we hypothesized that there would be a strong link', or something of this sort- need to improve the phrasing.

Author response: we will modify this accordingly in the revised version.

REVIEWER COMMENT 7 by Referee #2: Line 7 forward. This paragraph is not very clear – why will a long-term sampling strategy in particular help resolve the question of when and under what conditions cell-free EE is important? One could easily imagine that a detailed biochemical investigation of the nature and structure and dynamics of

cell-free enzymes is what is needed (tho this of course is technically very challenging, and is a different question than the one examined here.) The whole paragraph needs more focus. As it stands currently, this hypothesis is not very well phrased. At minimum, add another sentence or two: is this hypothesis based on rates of molecular motion, on rates of diffusion of enzymes and proteases, on an idea of the structural basis of enzymes that are produced under cold vs warm conditions, on kinetics of degradation/inactivation reactions?

Author response: we will add another sentence/s as suggested by the reviewer. We will specify that our hypothesis is based on the fact that higher lifetimes of EEA have been found in cold compared to warm waters, suggesting that an overall low metabolic rates of microbes would be more prone to higher percentages of dissolved EEA because the degradation of the enzymes is reduced under lower temperatures. We will also specify that, despite the field long-term effort required, seasonal analysis are good to help shed light into this question because temperature strongly changes seasonally and field studies are usually more representative of what occurs in nature than experimental manipulations.

REVIEWER COMMENT 8 by Referee #2: Pg. 4 Were killed controls (autoclaved water) subtracted from the measured values? Are other data (salinity, for example) available for this site? How variable are cell counts, or bacterial productivity, or DOM or POM or other biogeochemical parameters for this site?

Author response: Subsamples without substrate additions served as blanks to determine the background fluorescence of the samples. This is agreement with previous reports showing insignificant abiotic hydrolysis of the substrates (e.g. Hoppe HG (1993) Use of fluorogenic model substrates for extracellular enzyme activity (EEA) measurement of bacteria. Handbook of methods in aquatic microbial ecology: 423-431; Azúa I, Uanue M, Ayo B, Arrtolozaga I, Arrieta JM, et al. (2003) Influence of organic matter quality in the cleavage of polymers by marine bacterial communities. Journal of Plankton Research 25: 1451-1460; Unanue M, Ayo B, Agis M, Slezak D, Herndl GJ, et

al. (1999) Ectoenzymatic activity and uptake of monomers in marine bacterioplankton described by a biphasic kinetic model. Microb Ecol 37: 36-48). We did not include all the details of the methods because they have been already explained in previous works cited in the text, but we will include this information in the revised version. As mentioned in comment 2, we will also include bacterial production and DOM data and comment on it in the revised version of the ms.

REVIEWER COMMENT 9 by Referee #2: Pg 5: The data presentation and discussion on pg 5 is not very satisfactory. The BGase and LAPase plots do appear to track one another to some extent, but having the APase activity on the same plot and scale makes something of a mess. Try plotting with a secondary Y axis for APase. More critically, it is difficult to figure out what connections there might be: as the text says, there are sometimes –but not always- coincidental peaks in these different activities, but this pattern is not very strong.

Author response: We will plot it using a secondary axis Y for APase as suggested, but we think that it is valuable to plot all the enzymes together because it is relevant to see how the biggest peaks in BGase and LAPase coincide with peaks in APase as well.

REVIEWER COMMENT 10 by Referee #2: Moreover, the authors assert that BGase and LAPase show relative pro- tein/polysaccharide degradation ratios. Note that more recent work has demonstrated that this perspective is likely limited. Multiple investi- gations of a range of peptidases substrates (see in particular the work of Obayashi & Suzuki Obayashi, Y. and S. Suzuki (2005). "Proteolytic enzymes in coastal surface sea- water: Significant activity of en- dopeptidases and exopeptidases." Limnol. Oceanogr. 50: 722-726. Obayashi, Y. and S. Suzuki (2008). "Occurrence of exo- and endopep- tidases in dissolved and partic- ulate fractions of coastal seawater." Aq. Microb. Ecol. 50: 231-237. Bong, C. W., Y. Obayashi and S. Suzuki (2013). "Succession of protease activity in seawater and bacterial isolates during starvation in a mesocosm experiment." Aq. Microb. Ecol. 69: 33-46) which demonstrates that LAPase activity levels vary in a manner not indicative of other peptidase activities. See also the recent work of Steen

et al. (2015; "Sub- strate specificity of aquatic extracellular peptidases assessed by competitive inhibition assays using synthetic substrates" AME 75: 271-281), who investigate the specificity of LAPase-hydrolyzing enzymes, and comment that "In some studies, Vmax for Leu-AMC hydrolysis is interpreted as a quantitative proxy of the total peptidolytic potential of a community (Kellogg et al. 2011). The results presented here show that approach to be invalid." Furthermore, the work of Arnosti (2011), which is cited in the ms, discusses the issue of the broad and variable spectrum of polysaccharide hydrolases present in the ocean; measurements of BGase do not present a complete or representative picture.

Author response: We will discuss about the limitations behind the BGase:LAPase in the text, including the citations referred to by the reviewer.

REVIEWER COMMENT 11 by Referee #2: Note in addition that the peaks in BGase/LAPase ratios (which are not all that convinc- ing..this plot shows a general increasing trend over the timecourse of the study, and the single large peak is driven by 2 measurements of BGase activity) do not in fact coincide with the chl a max, since the peak chl a in Fig 1B occurs in ca August and then in May, whereas Fig 2b shows peaks in June and August.

Author response: In the study area, there is always an increases in chlorophyll found between April-May (diatom and dinoflagellate spring bloom) and July-September (cyanobacterial summer bloom), as shown for example in our detailed previous paper (Lindh et al. 2015, Disentangling seasonal bacterioplankton population dynamics by high-frequency sampling: 17 (7), 2459–2476). The peaks in BGase:LAPase ratio are not exactly at the time of the blooms but are just following after the blooms (after a time lag). This lag makes sense because first comes primary productivity and then heterotrophic degradation, but not usually exactly at the same time. We will explain this better in the revised version of the ms.

REVIEWER COMMENT 12 by Referee #2: The comments on lines 19-22 on different

phytoplankton blooms and enzyme activities is simply not very convincing: APase activity might be directly connected to phytoplank- ton, given the presence of APase in some phytoplankton, but the BGase and LAPase connection would be more distant, given that heterotrophic prokaryotes are the sources of these enzymes. The discussion of all of these data in fact leave the reader with the impression that the authors don't quite know what to do with this long time series. More digging in the literature, discussion with other colleagues who have data from this site might help, but this is currently a very weak part of the manuscript.

Author response: We probably did not explain the potential link between different phytoplankton blooms and enzyme activities sufficiently well. We partly agree with the reviewer suggesting that Apase might be directly connected to phytoplankton but much less BGase and LAPAse due to the presence of Apase in some phytoplankton. However, we also think that different groups of phytoplankton can release different types of organic carbon compounds, which would likely select for different bacterioplankton groups/enzymes (Pinhassi et al. 2004 Changes in Bacterioplankton Composition under Different Phytoplankton Regimens, 70 (11), 6753-6766), further suggesting a potential link between phytoplankton groups and enzyme activities. The main aim of this study is to focus on the changes in the % of dissolved EEA, since this is the novel part of it, and there are many more studies on the bulk EEA. That is why we only tried to briefly describe what happens in the bulk EEA without going too much into detail just to set the scene before talking about the dynamics of the percentage of dissolved EEA. We will explain better the potential link between different phytoplankton groups and changes in the enzyme activities and in the revised version of the ms.

REVIEWER COMMENT 13 by Referee #2: Pg 6 The high fraction of dissolved activity is indeed an interesting observation. The authors need to discuss here some ideas about why this might be - certainly others have also at times high cell-free activities, but these activities are high for most of the annual cycles. Moreover, the changes in the cell-free activities for example between March and May, when the data spikes up

and down with considerable frequency, would allow for some discussion of lifetimes (for activity to drop this much, some of the en- zymes must either be removed, or they are advected away, and one is sampling a different patch of water; to evaluate this, more data on the study site and the dynamics of the water masses is needed.)

Author response: We appreciate the suggestions by the reviewer about the changes observed between March and May. We think that besides what the reviewer suggests, the variability observed in March-June could also be related to the phytoplankton bloom (also observed as an increase variability in APase, and an increase in the BGase:LAPase ratio), and the rapid succession observed in different phytoplankton taxa during those months in this study site ((Lindh et al. 2015, Disentangling seasonal bacterioplankton population dynamics by high-frequency sampling: 17 (7), 2459–2476). We will extend our discussion on this topic in the revised version.

REVIEWER COMMENT 14 by Referee #2: As others have doubtless pointed out, correlation is not causation: there is a statistical correlation between T and cell-free activity, but this is a long way from a coherent or plausible explanation. To what extent do cell counts, bacterial productivity, or microbial community composition change at this site on seasonal scales? What is the physical oceanography of this setting –what water masses are sampled? It is extremely likely that the nature and types of enzymes that hydrolyze leu-MCA, B-MUF, and alkaline phosphatase are likely quite different under different conditions of productivity, and with different seasons, all factors that are correlated with differences in microbial community composition and activity. See for example Arrieta & Herndl (2002; "Changes in $\beta$- glucosidase diversity during a coastal phytoplankton bloom." Limnol Oceanogr 47: 594−599). In any case, drawing parallels between community activities in the Arctic and the Baltic needs also to consider the differences between permanently cold and temperate environments, in terms of community composition and activities. With all of these issues that should be addressed, extending the discussion here to questions of global warming is much too far a reach.

Author response: We agree with the reviewer that correlations are not evidence for

causal relationships, but also think that we need to show when significant correlations were found. A significant correlation together with a sensible hypothesis supporting it (based on previous results) motivates statistical analyses the use of significant correlations. As mentioned in several previous comments, we will include in the revised version the data on cell counts and bacterial productivity, as well as include information on our previous study of the study region showing a detailed seasonal study of microbial community composition (Lindh et al. 2015). Despite the fact that so many different factors can affect the types of enzymes present, a clear and robust seasonal pattern observed in the % of dissolved EEA, what gives even more value to this pattern observed in this study, and probably reinforces the idea that an integrating parameter (like temperature) that affects many other factors might be the main responsible in the regulation of the dissolved enzymes. This is also supported by the the fact that longer lifetimes have been reported in the Artic and deep Atlantic than surface waters, further indicating that the differences in temperature and all the associated changes that come with it (e.g. changes in heterotrophic rates, community composition, etc.), might be the main driver of the % dissolved EEA. If temperature is affecting the proportion of dissolved EEA we believe that at least mentioning a potential link with global warming is conceivable.

---

## Author Response (AR1)

**Point-by-point response and associated relevant changes made**

**Author response to Reviewer #1**
We thank the reviewer for the constructive comments on this manuscript. We have taken them on board and our responses to reviewer comments, including the modifications done to the manuscript, are detailed in the following:

REVIEWER COMMENT 1 by Referee #1:
General comment The article by Baltar et al. shows a strong relationship between the proportion of dissolved extracellular enzymatic activities and seasonal variations (temperature) occurring at a coastal site in the Baltic Sea. The topic of the manuscript is very important, and not fully understood yet. It is a clear and straightforward document that describes a valid experimental design.

Author response: We appreciate the positive comments of the reviewer

REVIEWER COMMENT 2 by Referee #1:
My only concern regards the lenght of the paper. In particular, the authors focus only on one important environmental factor: temperature. In order to make the article's conclusions sound (e.g. the potential effect of global warming on organic matter degradation performed by cell-free enzymes) some experimental manipulations could have been performed. Alternatively, the effect of factors other than temperature could have been considered (and eventually discarded if not significant). For these reasons I suggest to improve the analysis with other environmental factors (if available). Alternatively, the manuscript should be considered for a 'short note' type of paper. Since I'm aware that BG does not include 'short notes' as potential article types, an option could be the 'Ideas and perspectives' type, within which the article could perfectly fit in its current form with a slight modification of the title (just as a suggestion, it could become something like: "temperature affects the activity of cell-free extracellular enzymes: a seasonal case study in the Baltic Sea).

Author response: We have included other factors than temperature and light (e.g. nutrients, chlorophyll-a, salinity, DOC, bacterial abundance, bacterial production) in the analysis (new Table 1) and we have revised manuscript text accordingly (p.7, l.30 to p. 8, l. 2; p. 8, 10-15).
We appreciate the title suggested by the reviewer, and have modified the title inspired on the reviewer suggestion. It now reads: "Cell-free extracellular enzymatic activity is linked to seasonal temperature changes: a case study in the Baltic Sea".

REVIEWER COMMENT 3 by Referee #1:
Specific comments:
Page 1 line 27-29: This conclusive sentence is very strong: although potentially true it would deserve further investigations (see General comment)

Author response: We have rewritten this sentence to make it less strong and to include a suggestion for the need for further investigations in the revised version (p. 1, l. 28 to p. 2, l. 2). The sentence now reads: " This might also suggest a potential effect of global warming on the hydrolysis of organic matter via a reduction of the contribution of of cell-free enzymes to the bulk hydrolytic activity, and call for the need of further research to confirm it."

REVIEWER COMMENT 4 by Referee #1:
Page 2 line 3: although heterotrophic prokaryotes are much of the story, I would not ascribe only to them the pivotal DOM reworking role, especially when EEA are involved (exoenzymes from cyanobacteria can be even more efficient). I'd be more cautious just saying 'prokary- otes'. Page 2 line 15: extra ')'

Author response: We have modified this accordingly, and the sentences now read (p. 2, l. 5): " Prokaryotes play a central role…". We have also deleted the extra ')'.

REVIEWER COMMENT 5 by Referee #1:
Page 3 line 25: "Temperature was measured on site" How? Was water collected and temperature measured on a aliquot by means of a thermometer? Anyhow this should be indicated.

Author response: We have included this information in the revised version (p. 4, l. 6-8), which reads as follows: " Temperature was measured on site through thermometer placed in the Ruttner sampler, and the water was transported to the laboratory in acid-washed Milli-Q-rinsed polycarbonate bottles within 1 h.".

REVIEWER COMMENT 6 by Referee #1:
Page 6 line 8: I am afraid this (temperature and light) is a little restrictive. pH is an important factor in enzyme activity, especially in extracellular ones. What is the pH variability in the 'low-salinity' high-chlorophyll Baltic sea? I suspect that in the sampling area pH variations might be broader that those tested in many ocean acidification experiments for which many references are available. Substrate concentration and composition also affect enzyme activity with or without links to the metabolic state of the source organism (Arnosti 2011) Catalytic elements have also been shown to drive (to some extent) hydrolysis rates of at least LAPase (Fukuda et al., 2000). These aspects (especially pH and substrate) should be mentioned.

Author response: The reviewer is correct on pointing to those factors (pH and substrate) as potentially affecting bulk EEA. However, in that paragraph we are describing the potential factors explaining/affecting the patterns in "dissolved EEA" (that is the central point of the study), but not in the bulk/total EEA. We are not aware of studies indicating that substrate or pH affect the proportion of dissolved EEA in seawater, so we cannot add anything about it. Nevertheless, to improve the interpretation of factors potentially controlling the proportion of dissolved EEA, we have now included other measured variables as well (see response to reviewer comment 2), although we did not measure pH (pH *in situ* is extremely challenging to measure with the required accuracy for in situ relevance); but we hope the reviewer will agree that the strongest variations in pH would be caused by autotrophic and heterotrophic activity, wherefore our inclusion of phytoplankton biomass

[chlorophyll-a] and bacterial heterotrophic activity, which are linked to fluctuations in pH, have relevance).

REVIEWER COMMENT 7 by Referee #1:
Page 10 line 30: heterotrophic MARINE flagellates

Author response: Fixed now.

REVIEWER COMMENT 8 by Referee #1:
Figure 2, upper panel: error bars should be shown

Author response: We have now included the error bars in Figure 2.

**Author response to Reviewer #2**
We thank the reviewer for the constructive comments on this manuscript. We have taken them on board and our responses to reviewer comments including the modifications done to the manuscript:

REVIEWER COMMENT 1 by Referee #2:
General comments: The authors attempt to address an interesting and important problem, with a time series of measurements of potential enzyme activities, including the fraction that is cell free. These measurements are a good start, but the data lack context, and the framework which the authors build does not adequately reflect current understanding of microbial community activities and dynamics.

Author response: Our main objective (and novel contribution) was to focus on the factors controlling the proportion of dissolved EEA, since there are many other reports in the literature on the factors controlling the bulk EEA. To increase the background and the depth of our analyses of the temporal patterns of the proportion of dissolved EEA, we have included several additional measured variables (concentration of nitrate, phosphate ammonium, chlorophyll-a and DOC, salinity, bacterial abundance, bacterial production) and built a new table (Table 2) showing these results (see response to reviewer 1 comment 2). We have also adjusted the framework according to the reviewer's comments (see below).

REVIEWER COMMENT 2 by Referee #2:
Are other data in addition to T and chl a available? Salinity, DOC, circulation, other physical or chemical data for this site? A multitude of factors can affect potential enzyme activities, as well as the fraction of activity that is cell-free. Are cell counts avail- able, or are there other studies done at the same location that provide a more in-depth background on the microbial ecology of the system? In particular, the Baltic has a very strong N-S salinity gradient that is linked with both compositional and functional changes in microbial communities (e.g. Dupont et al. (2014). "Functional tradeoffs underpin salinity-driven divergence in microbial community composition."

PLOS one 9: 89549). It would be helpful to have some sense of microbial dynamics at this site; the T and chl a data are not really a sufficient context.
In its current form, this manuscript lacks sufficient significance and quality and presen- tation; it needs a lot of work.

Author response: Yes, we have other data available, and we have included those in the revised version (new Table 1) (please see response to reviewer 1 comment 2 for details).

REVIEWER COMMENT 3 by Referee #2:
Specific comments: Pg 2 lines 11, 12 Note that Arnosti (2011) is cited improperly ("This is why the activity of extracellular enzymatic activity (EEA) has been recognized as the rate-limiting step in organic matter degradation (Arnosti, 2011)." Arnosti (2011) does not state that EEA is the rate limiting step: that reference points out that EEA is the initial step in organic matter degradation, and discusses this idea at great length. These statements are not the same!

Author response: We are sorry for this misunderstanding, we have changed "rate-limiting" for "initial" (p. 2, l. 13).

REVIEWER COMMENT 4 by Referee #2:
Lines 18, 19: earlier reports of cell-free activity, including Karner & Rassoulzadegan (1995), Obayashi & Suzuki (2008) and Keith & Arnosti (2001).
Note that one of the key points of the current ms - that enzyme activities may be decoupled from the producing organisms, and may contribute greatly to carbon cycling - is explicitly also discussed in Arnosti (2011), and summarized as one of the key points in that manuscript.

Author response: We have included those references (Karner & Rassoulzadegan (1995) and Obayashi & Suzuki (2008) and Keith & Arnosti (2001)) at the end of that statement (p. 2, l. 20-21). We have also included references of the previous papers where the decoupling of EEA from the producing organisms has been suggested before ((Arnosti, 2011) and Baltar et al., 2010) (p. 2, l. 25-26)).

REVIEWER COMMENT 5 by Referee #2:
Lines 31 forward: Steen & Arnosti (2011) investigated cell-free enzyme lifetime in Arctic surface waters

Author response: we have included that reference in that specific statement (p. 3, l. 2-3).

REVIEWER COMMENT 6 by Referee #2:
Pg. 3: Line 7, delete 'the' before 'cell-free' Line 8, delete 'what are' before 'the factors' Line 8, this thought is incomplete. Do you mean 'little is known about the factors that control changes in the proportion of total EEA that is dissolved' ? Line 9, move 'is needed' to the end of the sentence Line 11: move the information about the specific enzyme activities assayed into another sentence following this one. Line 14: change 'decipher what factors' to 'decipher the factors that control' Line 16, change to 'we hypothesized that there would be a strong link', or something of this sort- need to improve the phrasing.

Author response: we have modified these points following exactly the reviewer's suggestions (now on p. 3, l. 11-22).

REVIEWER COMMENT 7 by Referee #2:
Line 7 forward. This paragraph is not very clear – why will a long-term sampling strategy in particular help resolve the question of when and under what conditions cell-free EE is important? One could easily imagine that a detailed biochemical investigation of the nature and structure and dynamics of cell-free enzymes is what is needed (tho this of course is technically very challenging, and is a different question than the one examined here.) The whole paragraph needs more focus.
As it stands currently, this hypothesis is not very well phrased. At minimum, add another sentence or two: is this hypothesis based on rates of molecular motion, on rates of diffusion of enzymes and proteases, on an idea of the structural basis of enzymes that are produced under cold vs warm conditions, on kinetics of degradation/inactivation reactions?

Author response: We have now specified that despite the intrinsic effort required to carry out a long-term seasonal analysis on natural samples collected in the field, they are valuable to help shed light into this question because temperature strongly changes seasonally and field studies are usually more representative of what occurs in nature than experimental manipulations (p. 3, l. 11-15).
We have also added another sentence/s as suggested by the reviewer specifying that our hypothesis is based on the fact that higher lifetimes of EEAs have been found in cold compared to warm waters, suggesting that an overall low metabolic rate of microbes would result in higher percentages of dissolved EEA because the degradation of the enzymes is reduced under lower temperatures (p. 3, l. 24-28).

REVIEWER COMMENT 8 by Referee #2:
Pg. 4 Were killed controls (autoclaved water) subtracted from the measured values? Are other data (salinity, for example) available for this site? How variable are cell counts, or bacterial productivity, or DOM or POM or other biogeochemical parameters for this site?

Author response: Subsamples without substrate additions served as blanks to determine the background fluorescence of the samples. This is agreement with previous reports showing insignificant abiotic hydrolysis of the substrates (e.g. Hoppe HG (1993) Use of fluorogenic model substrates for extracellular enzyme activity (EEA) measurement of bacteria. Handbook of methods in aquatic microbial ecology: 423-431; Azúa I, Uanue M, Ayo B, Arrtolozaga I, Arrieta JM, et al. (2003) Influence of organic matter quality in the cleavage of polymers by marine bacterial communities. Journal of Plankton Research 25: 1451-1460; Unanue M, Ayo B, Agis M, Slezak D, Herndl GJ, et al. (1999) Ectoenzymatic activity and uptake of monomers in marine bacterioplankton described by a biphasic kinetic model. Microb

Ecol 37: 36-48). We have included this information in the revised version (see p. 5, l. 1-5).

We also included bacterial production and DOM data (as well as other variables) and comment on it in the revised version of the ms (see response to comment 2, and response to reviewer 1 comment 2 for details).

REVIEWER COMMENT 9 by Referee #2:
Pg 5: The data presentation and discussion on pg 5 is not very satisfactory. The BGase and LAPase plots do appear to track one another to some extent, but having the APase activity on the same plot and scale makes something of a mess. Try plotting with a secondary Y axis for APase. More critically, it is difficult to figure out what connections there might be: as the text says, there are sometimes –but not always- coincidental peaks in these different activities, but this pattern is not very strong.

Author response: We have followed the suggestion by the reviewer and redone Figure 2 by plotting with a secondary Y-axis for APase. Now it is much clearer to see the strong connection (including coincidental peaks) that exists between all the different EEAs. We have also included a correlation analysis of the different bulk EEAs, explaining better their relations (p. 6, l. 7-12).

REVIEWER COMMENT 10 by Referee #2:
Moreover, the authors assert that BGase and LAPase show relative pro-tein/polysaccharide degradation ratios. Note that more recent work has demonstrated that this perspective is likely limited. Multiple investigations of a range of peptidases substrates (see in particular the work of Obayashi & Suzuki Obayashi, Y. and S. Suzuki (2005). "Proteolytic enzymes in coastal surface seawater: Significant activity of en- dopeptidases and exopeptidases." Limnol. Oceanogr. 50: 722-726. Obayashi, Y. and S. Suzuki (2008). "Occurrence of exo- and endopeptidases in dissolved and partic- ulate fractions of coastal seawater." Aq. Microb. Ecol. 50: 231-237. Bong, C. W., Y. Obayashi and S. Suzuki (2013). "Succession of protease activity in seawater and bacterial isolates during starvation in a mesocosm experiment." Aq. Microb. Ecol. 69: 33-46) which demonstrates that LAPase activity levels vary in a manner not indicative of other peptidase activities. See also the recent work of Steen et al. (2015; "Sub- strate specificity of aquatic extracellular peptidases assessed by competitive inhibition assays using synthetic substrates" AME 75: 271-281), who investigate the specificity of LAPase-hydrolyzing enzymes, and comment that "In some studies, Vmax for Leu-AMC hydrolysis is interpreted as a quantitative proxy of the total peptidolytic potential of a community (Kellogg et al. 2011). The results presented here show that approach to be invalid." Furthermore, the work of Arnosti (2011), which is cited in the ms, discusses the issue of the broad and variable spectrum of polysaccharide hydrolases present in the ocean; measurements of BGase do not present a complete or representative picture.

Author response: We have included a discussion about the limitations behind the BGase:LAPase, including all the citations referred to by the reviewer (p. 6, l. 18-26).

REVIEWER COMMENT 11 by Referee #2:
Note in addition that the peaks in BGase/LAPase ratios (which are not all that convinc- ing..this plot shows a general increasing trend over the timecourse of the study, and the single large peak is driven by 2 measurements of BGase activity) do not in fact coincide with the chl a max, since the peak chl a in Fig 1B occurs in ca August and then in May, whereas Fig 2b shows peaks in June and August.

Author response: In the study area, there is always an increase in chlorophyll found between April-May (diatom and dinoflagellate spring bloom) and July-September (cyanobacterial summer bloom), as shown for example in our previous paper (Lindh et al. 2015, Disentangling seasonal bacterioplankton population dynamics by high-frequency sampling. Environmental Microbiology: 17 (7), 2459–2476). The peaks in BGase:LAPase ratio are not exactly at the time of the blooms but are following just after the blooms (with a time lag). This lag makes sense because first comes primary productivity and then heterotrophic degradation, but they do not usually occur exactly at the same time. We have done our best to explain this better in the revised version of the ms (p. 6, l. 12-16).

REVIEWER COMMENT 12 by Referee #2:
The comments on lines 19-22 on different phytoplankton blooms and enzyme activities is simply not very convincing: APase activity might be directly connected to phytoplank- ton, given the presence of APase in some phytoplankton, but the BGase and LAPase connection would be more distant, given that heterotrophic prokaryotes are the sources of these enzymes.
The discussion of all of these data in fact leave the reader with the impression that the authors don't quite know what to do with this long time series. More digging in the literature, discussion with other colleagues who have data from this site might help, but this is currently a very weak part of the manuscript.

Author response: We partly agree with the reviewer suggesting that Apase might be directly connected to phytoplankton but much that the connection between phytoplankton and BGase and LAPAse would be less evident due to the presence of APase in some phytoplankton. However, we also think that different groups of phytoplankton can release different types of organic carbon compounds, which would likely select for different bacterioplankton groups/enzymes (Pinhassi et al. 2004 Changes in Bacterioplankton Composition under Different Phytoplankton Regimens, 70 (11), 6753-6766), further suggesting a potential link between phytoplankton groups and enzyme activities. In the revised version of the manuscript we now attempt to clearly explained better this potential link between phytoplankton community structure and changes in EEA (p. 6, l. 26-32).

REVIEWER COMMENT 13 by Referee #2:
Pg 6 The high fraction of dissolved activity is indeed an interesting observation. The authors need to discuss here some ideas about why this might be - certainly others have also at times high cell-free activities, but these activities are high for most of the annual cycles. Moreover, the changes in the cell-free activities for example between March and May, when the data spikes up and down with considerable frequency, would allow for some discussion of lifetimes (for activity to drop this much, some of the en- zymes must either be removed, or they are advected away, and one is sampling a different patch of water; to evaluate this, more data on the study site and the dynamics of the water masses is needed.)

Author response: We think that the variability observed in March-June could also be related to the phytoplankton bloom (also observed as an increase variability in APase, and an increase in the BGase:LAPase ratio), and the rapid succession we observed in different phytoplankton taxa during those months in this study site ((Lindh et al. 2015, Disentangling seasonal bacterioplankton population dynamics by high-frequency sampling: Environmental Microbiology: 17 (7), 2459–2476). We have extended our discussion on this topic, including our best potential interpretation of the ups and downs observed in the proportion of dissolved EEA between March and May in the revised version (p. 7, l. 13-18), which we hope helps the reader interpret our data.

REVIEWER COMMENT 14 by Referee #2:
As others have doubtless pointed out, correlation is not causation: there is a statistical correlation between T and cell-free activity, but this is a long way from a coherent or plausible explanation. To what extent do cell counts, bacterial productivity, or microbial community composition change at this site on seasonal scales? What is the physical oceanography of this setting –what water masses are sampled? It is extremely likely that the nature and types of enzymes that hydrolyze leu-MCA, B-MUF, and alkaline phosphatase are likely quite different under different conditions of productivity, and with different seasons, all factors that are correlated with differences in microbial community composition and activity. See for example Arrieta & Herndl (2002; "Changes in β- glucosidase diversity during a coastal phytoplankton bloom." Limnol Oceanogr 47: 594−599). In any case, drawing parallels between community activities in the Arctic and the Baltic needs also to consider the differences between permanently cold and temperate environments, in terms of community composition and activities.
With all of these issues that should be addressed, extending the discussion here to questions of global warming is much too far a reach.

Author response: We agree with the reviewer that correlations are not evidence for causal relationships, but also think that we need to show when significant correlations were found. A significant correlation together with a sensible hypothesis supporting it (based on previous results) supports the utilization of significant correlations. We have now included in the revised version, among others, the data on cell counts and bacterial productivity (see response to reviewer comment 2), as well as included information on our previous study of the study region showing a detailed seasonal study of microbial community composition (Lindh et al. 2015).
Despite the fact that so many different factors can affect the types of enzymes present, a clear and robust seasonal pattern was observed in the % of dissolved EEA, which we consider gives even more value to this pattern observed in this study, and probably reinforces the idea that an integrating variable (like temperature) that affects many other factors might be the main responsible in the regulation of the dissolved enzymes. This is also supported by the fact that longer lifetimes have been reported in the Artic and deep Atlantic than surface waters, further indicating that the differences in temperature and all the associated changes that come with it (e.g. changes in heterotrophic rates, community composition, etc.), might be the main driver of the %

dissolved EEA. This is further supported now by our new analysis done which shows a significant negative correlation between the proportion of dissolved EEA (BGase and LAPase) and the heterotrophic metabolic activities (bacterial production). We have now included discussion on this topic (p. 8, l. 10-15)

If temperature is the variable most strongly affecting the proportion of dissolved EEA (as suggested by our results), we believe that at least mentioning a potential link with global warming is appropriate. Nevertheless, we have rewritten these statements (on the link to global warming), at the end of the abstract and at the end of the Discussion, to make it less strong (p. 1, l. 28 to p. 2, l. 2 and p. 8, l. 20-21).

END OF REVISION

[revised manuscript text omitted]